

# Drude weight for the Lieb-Liniger Bose gas

Benjamin Doyon[1] and Herbert Spohn[2]

**1** Department of Mathematics, King's College London, Strand, London WC2R 2LS, U.K.
**2** Physik Department and Zentrum Mathematik, Technische Universität München, Boltzmannstr. 3, 85748 Garching, Germany

## Abstract

Based on the method of hydrodynamic projections we derive a concise formula for the Drude weight of the repulsive Lieb-Liniger $\delta$-Bose gas. Our formula contains only quantities which are obtainable from the thermodynamic Bethe ansatz. The Drude weight is an infinite-dimensional matrix, or bilinear functional: it is bilinear in the currents, and each current may refer to a general linear combination of the conserved charges of the model. As a by-product we obtain the dynamical two-point correlation functions involving charge and current densities at small wavelengths and long times, and in addition the scaled covariance matrix of charge transfer. We expect that our formulas extend to other integrable quantum models.



# 1   Introduction

The hydrodynamic description of fluids is based on the notion of local equilibrium: in a cell, containing many atoms but still very small on the macroscopic scale, the fluid is in thermal equilibrium. The local equilibrium parameters change slowly in space-time and are governed by an autonomous system of evolution equations. This gives a very powerful method to study non-equilibrium systems and large-scale response functions. To carry out such a program one first has to identify the local conservation laws. For a fluid in physical space there are five: mass, momentum, and energy. To lowest order in the spatial gradients one obtains time-reversible evolution equations (Euler equations) and to second order dissipative corrections (Navier-Stokes equations) [1,2]. Recently [3,4] there has been a lot of progress in generalizing the hydrodynamic picture to integrable systems in one space dimension, for which the number of conserved fields is extensive. A priori, it is not so clear whether the standard heuristic survives such a drastic extension. But the recent studies are encouraging. Furthermore, one has the worked out example of a hard rod fluid [5–7], for which the number of particles at given velocity is conserved. For the hard rod fluid the first order Euler-type equations are known and also their dissipative corrections. In particular, it is proved that local equilibrium is maintained throughout space-time on the macroscopic scale.

For quantum integrable models, even to write down generalized hydrodynamics might be difficult, let along to solve it. One has to know not only all conserved fields, which usually come together with integrability, but also their associated currents. From the conserved fields one constructs the generalized Gibbs ensemble (GGE) [8–10], which contains an infinite number of "chemical potentials". For the Euler-type equation the average fields and currents are required in GGEs. In principle these are available for every Bethe-ansatz integrable model or integrable quantum field theory [3, 4], the only information necessary being the spectrum of Bethe (or asymptotic) particles, their energies, and their scattering phases. These building blocks have been explicitly studied in particular for the XXZ spin chain [4], and for the sinh-Gordon model and its non-relativistic limit the Lieb-Liniger $\delta$-Bose gas [3], the latter being of interest in our note. For these models the Euler-type conservation equations have been derived, including force terms produced by external fields such as those from confining potentials [11]. Their tentative dissipative corrections are not yet known, see the numerical study [12].

The Euler-type equations can be numerically solved and compared to results on quantum evolutions in a variety of ways. There are integral equations for the general initial value problem (without external force) [13], which efficiently produce exact solutions by iteration. Using different methods, the collision of two clouds of particles in the Lieb-Liniger model are simulated, finding agreement with DMRG numerics [14]. In the limit of zero temperature, the equations reduce to a finite family of hydrodynamic conservation laws [15]. Thereby the evolution of density waves in the Lieb-Liniger model, with or without confining potentials, was analyzed, observing agreement with the exact quantum evolution based on the Bethe ansatz. An efficient molecular dynamics scheme has been proposed [16], which also accounts for external forces in the Lieb-Liniger model. For the problem of domain wall initial states [17–19], in which initially the GGE chemical potentials are constant except for a possible jump at the origin, exact analytic solutions have been obtained using generalized hydrodynamics [3,4,20].

In this note, we concentrate on stationary, homogeneous states, hence GGEs, and we explain how to compute the exact Drude weight and related quantities for the Lieb-Liniger model in the repulsive regime in arbitrary GGEs. Since the formalism is general, the method also applies, given the particle spectrum and scattering, to other integrable models, and, conjecturally and with appropriate modifications, to classical soliton-like gases [16] and integrable classical field theory (perhaps using the results of [21]). The derivation makes use of generalized hydrodynamics by combining it with hydrodynamic projection methods [22,23]. As a prelimi-

nary step we remind the reader in Section 2 how, for a finite number of conservation laws, the Drude weight is computed using hydrodynamic projections. We emphasize that it is important to regard all conserved quantities on equal footing, and thus the Drude weight as a matrix. By looking at a particular matrix element, one might miss the global structure. To prepare for the general case, we then consider in Section 3 the Drude weight for a hard rod fluid [5], see also [6,7]. In this case the Drude weight is an infinite-dimensional matrix, or bilinear functional. All is well-known material, but arranged in such a way as to emphasize the analogy with the Lieb-Liniger model.

Our main results, concerning the Lieb-Liniger model and more general integrable models, are reported in Section 4, but we list already here the main identities:

$$\int dx \, \langle \mathsf{q}_i(x,0)\mathsf{q}_j(0,0)\rangle^c = \int d\theta \, \rho_p(\theta)(1-\sigma n(\theta))h_i^{dr}(\theta)h_j^{dr}(\theta), \tag{1.1}$$

$$\int dx \, \langle \mathsf{j}_i(x,0)\mathsf{q}_j(0,0)\rangle^c = \int d\theta \, \rho_p(\theta)(1-\sigma n(\theta))v^{eff}(\theta)h_i^{dr}(\theta)h_j^{dr}(\theta), \tag{1.2}$$

$$\lim_{t\to\infty}\int dx \, \langle \mathsf{j}_i(x,t)\mathsf{j}_j(0,0)\rangle^c = \int d\theta \, \rho_p(\theta)(1-\sigma n(\theta))v^{eff}(\theta)^2 h_i^{dr}(\theta)h_j^{dr}(\theta), \tag{1.3}$$

$$\int dt \, \langle \mathsf{j}_i(0,t)\mathsf{j}_j(0,0)\rangle^c = \int d\theta \, \rho_p(\theta)(1-\sigma n(\theta))|v^{eff}(\theta)|h_i^{dr}(\theta)h_j^{dr}(\theta), \tag{1.4}$$

where

$$\sigma = 1, -1, 0 \quad \text{for fermionic, bosonic, and classical gases respectively.}$$

The quantity $\rho_p$ is the density of particles per unit distance and per unit spectral parameter $\theta$, $n(\theta)$ is the usual occupation function of the (generalized) thermodynamic Bethe ansatz [24, 25] (or the classical free density [16, 20]), and $v^{eff}(\theta)$ is the effective velocity [3, 4, 26]. The superscript $^{dr}$ represents the dressing operation of the thermodynamic Bethe ansatz (see [3, 24]). On the left-hand side are GGE connected correlation functions.

In quantum systems, one takes $\mathsf{q}_i(x,t) = \hat{Q}_i(x,t)$ and $\mathsf{j}_i(x,t) = \hat{J}_i(x,t)$, respectively the $i$-th conserved charge density and its current. In this case, $h_i(\theta)$ is the one-particle eigenvalue, at spectral parameter $\theta$, of the associated conserved charge $\int dx \, \hat{Q}_i(x,0)$. For the one-dimensional classical fluid of hard rods, one identifies $\theta$ with the particle velocity $v$, and takes $\mathsf{q}_i(x,t) = \sum_\ell h_i(v_\ell)\delta(x-r_\ell)$ and $\mathsf{j}_i(x,t) = \sum_\ell h_i(v_\ell)\dot{r}_\ell\delta(x-r_\ell)$, respectively the conserved density and its associated current for a weight $h_i(v)$, where $r_\ell$ is the position and $v_\ell$ the velocity of the $\ell$-th particle in the fluid. Conjecturally, this would also hold for classical soliton-like gases.

For Bethe integrable models formula (1.1) is an immediate consequence of the (generalized) thermodynamic Bethe ansatz formalism, which provides the exact free energy. For the Lieb-Liniger model (1.1) restricted to the density has also been derived using form factors [27]. Formula (1.2) can be viewed as a consequence of the exact current "potential" obtained in [3]. The identity (1.3) is for the conventional Drude weight. Expression (1.4) gives the scaled covariance matrix of charge transfer, which we will refer to as "Drude self-weight" (this is called zero-frequency noise in mesoscopic physics [28]). Such scaled cumulants form an important part of the large-deviation theory for non-equilibrium transport [29,30]. We also obtain expressions for dynamical charge-charge, charge-current and current-current correlation functions at small wavelengths and large times. In the particular case of the Lieb-Liniger density-density correlation, our expression agrees with the result obtained from the form factor analysis [27].

The first expressions for particular components of the Drude weight at nonzero temperature in interacting integrable models were obtained in the context of spin chains. Expressions were found, by various methods, for the charge-charge Drude weight in the Hubbard

model [31], the spin-spin Drude weight in the XXZ chain [32], and the energy-energy Drude weight in the XXZ chain [33, 34]. In fact, the exact expression for the XXZ spin-spin Drude weight has been the subject of some debate. The situation was recently settled in a series of works [35–39]. In [38, 39], the spin-spin Drude weight was exactly evaluated by combining the hydrodynamic techniques of [3, 4] with a formula expressing it as a linear response of the non-equilibrium current to a change of driving potential [37, 40, 41]. Our method and expression are however new. Formula (1.3) confirms and generalizes the early results [31, 32]. As a consistency check, we show in Section 5 that it is reproduced in complete generality by the linear response calculation, thus further confirming that the numerical analysis of [37, 40, 41] agrees with these early results.

We also show Section 5 that our exact result for the Drude self-weight is reproduced by standard fluctuation relations [30, 42].

## 2 Models with a finite number of conservation laws

Before embarking on the Lieb-Liniger model, we briefly discuss the generic structure for models with a finite number, say $m$, of locally conserved fields. It is assumed that $m$ is already their maximal number. Thus for Galilean fluids in three dimensions $m = 5$, while for generic anharmonic chains and one-dimensional fluids $m = 3$. For quantum spin chains generically only the energy is conserved, hence $m = 1$. In our context one spatial dimension is in focus, and thus we only mention [43, App. A] and [44].

Microscopically we consider a one-dimensional system with $m$ locally conserved densities, $\mathsf{q}_j(x,t), j = 1, ..., m$ on space-time $(x,t) \in \mathbb{R}^2$, and their associated currents, $\mathsf{j}_j(x,t), j = 1, ..., m$, satisfying

$$\partial_t \mathsf{q}_j(x,t) + \partial_x \mathsf{j}_j(x,t) = 0. \tag{2.1}$$

Classically $\mathsf{q}_j$ and $\mathsf{j}_j$ may be seen as functions on phase space. The fields may also be seen as generated by a multi-species stochastic particle system. Quantum mechanically $\mathsf{q}_j$ and $\mathsf{j}_j$ would be operator fields indexed by $(x,t)$, with certain locality properties (see the brief discussion in the context of the Lieb-Liniger model). Their precise definition in terms of the underlying dynamics is not important at the present stage. Since $m$ is the maximal number of conservation laws, the microscopic system has an $m$-dimensional family of steady states, with distribution of the form $e^{-\sum_i \beta_i \int dx\, \mathsf{q}_i(x)}$. These states may be labelled by the Lagrange parameters $\beta_i$, or by the mean value of the conserved quantities. The time-stationary states are assumed to be invariant under spatial translations and the system is initialized in one of the time-stationary states. Hence the underlying dynamics is a space-time stationary random process, or a space-time invariant quantum field theory or quantum chain. We label the steady states by $\vec{u} \in \mathbb{R}^m$, and averages are denoted by $\langle \cdot \rangle_{\vec{u}}$. Since the steady states are completely specified by the averages of conserved densities $\mathsf{q}_j(x,t)$, we set by definition

$$\langle \vec{\mathsf{q}}(x,t) \rangle_{\vec{u}} = \vec{u}, \tag{2.2}$$

independent of $x, t$. For connected averages we use the notation $\langle ab \rangle_{\vec{u}}^{\mathrm{c}} = \langle ab \rangle_{\vec{u}} - \langle a \rangle_{\vec{u}} \langle b \rangle_{\vec{u}}$. The average currents are denoted by

$$\langle \vec{\mathsf{j}}(x,t) \rangle_{\vec{u}} = \vec{\mathsf{j}}(\vec{u}). \tag{2.3}$$

Any initial state which locally looks like one of the stationary states keeps this property under time evolution. In such situations, the state in space-time can be seen as locally stationary and homogeneous, and therefore completely characterized by a space-time function

$\vec{u}(x,t)$. This is the usual hydrodynamic approximation. In this approximation, the parameters characterizing the local state are governed by the system of macroscopic conservation laws,

$$\partial_t \vec{u}(x,t) + \partial_x \vec{j}(\vec{u}(x,t)) = 0. \tag{2.4}$$

In terms of the microscopic system, (2.4) is approximately valid on suitably large scales.

Let us go back to homogeneous and stationary states. From a statistical physics perspective, of particular interest is the correlator of the conserved fields in the stationary set-up,

$$S_{ij}(x,t) = \langle \mathfrak{q}_i(x,t)\mathfrak{q}_j(0,0)\rangle_{\vec{u}}^c, \tag{2.5}$$

with the fixed parameter $\vec{u}$ characterizing the statistically space-time homogeneous state. One should think of $S(x,t)$ as an $m \times m$ matrix. At such level of generality nothing can be said about the correlator. But on the hydrodynamic scale, which corresponds to large $x, t$, $S$ is linked to solutions of (2.4) linearized as $\vec{u} + \epsilon \vec{\phi}$ with constant $\vec{u}$.

First we write the linearized equation, obtained to first order in the small parameter $\epsilon$, as

$$\partial_t \vec{\phi}(x,t) + A\partial_x \vec{\phi}(x,t) = 0, \tag{2.6}$$

where

$$A_{ij}(\vec{u}) = \partial_{u_j} \mathfrak{j}_i(\vec{u}). \tag{2.7}$$

The matrix $A$ depends on $\vec{u}$ and acts only in component space. As further input we need the static covariance matrix

$$C_{ij} = \int \mathrm{d}x\, S_{ij}(x,t) = \int \mathrm{d}x\, S_{ij}(x,0) \tag{2.8}$$

and the field-current correlator

$$B_{ij}(\vec{u}) = \int \mathrm{d}x\, \langle \mathfrak{j}_i(x,0)\mathfrak{q}_j(0,0)\rangle_{\vec{u}}^c. \tag{2.9}$$

Note that, as $m \times m$-matrices,

$$B = AC. \tag{2.10}$$

This can be derived by the chain rule. Indeed, let $\beta_i$ be the conjugate potential to the conserved quantity $\int \mathrm{d}x\, \mathfrak{q}_j(x)$ in the homogeneous stationary state. This means that

$$\frac{\partial}{\partial \beta_i} \langle \mathfrak{a}(0,0)\rangle_{\vec{u}} = \int \mathrm{d}x\, \langle \mathfrak{q}_i(x,0)\mathfrak{a}(0,0)\rangle_{\vec{u}}^c \tag{2.11}$$

for any local field $\mathfrak{a}(x,t)$, such as $\mathfrak{q}_j(x,t)$ of $\mathfrak{j}_j(x,t)$. Hence, we find for instance $\partial_{\beta_i} \langle \mathfrak{q}_j \rangle_{\vec{u}} = C_{ij}$. Therefore in compressed notation, we have $B = \partial_{\vec{\beta}} \langle \vec{\mathfrak{j}} \rangle_{\vec{u}} = \partial_{\vec{\beta}} \langle \vec{\mathfrak{q}} \rangle \cdot \partial_{\vec{u}} \langle \vec{\mathfrak{j}} \rangle_{\vec{u}}$.

Then, one solves (2.6) with random initial conditions characterized by the static covariance $C$. This amounts to evaluating $\tilde{S}(x,t) = \lim_{\lambda \to \infty} \lambda S(\lambda x, \lambda t)$ by solving the evolution equation

$$\partial_t \tilde{S}(x,t) + \partial_x\big(A\tilde{S}(x,t)\big) = 0 \tag{2.12}$$

with initial condition $\tilde{S}(x,0) = \delta(x)C$, consistent with an exponential decay of $S(x,0)$, a general feature of models in one dimension at strictly positive temperatures. Therefore, in the hydrodynamic approximation, small $k$, large $t$, one has

$$\int \mathrm{d}x\, \mathrm{e}^{ikx} S(x,t) \simeq \mathrm{e}^{iktA}C. \tag{2.13}$$

Note in particular that changing variable to $x = \lambda x'$ and defining $k' = \lambda k$, after taking the limit $\lambda \to \infty$ with $k'$ fixed relation (2.13) holds for all values of $k'$, thus the inverse Fourier transform can be performed giving the correct initial condition.

Using only the conservation laws and space-time stationarity, in general one has the relation

$$AC = CA^{\mathrm{T}}, \qquad (2.14)$$

where $^{\mathrm{T}}$ denotes transpose. Of course, $C = C^{\mathrm{T}}$ by definition. But (2.10) together with (2.14) implies the less immediate symmetry

$$B = B^{\mathrm{T}}, \qquad (2.15)$$

which means in particular that the vector field $\vec{j}(\vec{u})$ is the gradient of a potential.

The conventional definition of the Drude weight is

$$D_{ij} = \lim_{t \to \infty} \frac{1}{t} \int_0^t \mathrm{d}t' \int \mathrm{d}x \, \langle \mathsf{j}_j(x,t')\mathsf{j}_i(0,0)\rangle_{\vec{u}}^{\mathrm{c}} = \lim_{t \to \infty} \int \mathrm{d}x \, \langle \mathsf{j}_j(x,t)\mathsf{j}_i(0,0)\rangle_{\vec{u}}^{\mathrm{c}}, \qquad (2.16)$$

provided the limit exists. It is convenient to view this expression as resulting from the inner product

$$\langle a|b\rangle = \int \mathrm{d}x \, \langle a(x)b(0)\rangle_{\vec{u}}^{\mathrm{c}} \qquad (2.17)$$

for general random fields, $a(x), b(x)$, which are statistically translation invariant in $x$. With respect to this scalar product, the conserved fields are in the time-invariant subspace. Assuming that the list of conserved fields is complete and the dynamics is sufficiently mixing,[1] one would expect that the time-invariant subspace is spanned by all the conserved total fields and hence the $t \to \infty$ limit is given by the projection onto this subspace (of course, with respect to the inner product (2.17)). In the statistical theory of fluids this step is called the hydrodynamic projection. With this reasoning, the long time limit in (2.16) is given by the projection onto the time-invariant subspace, which is given by

$$D_{ij} = \sum_{i',j'=1}^{m} \langle \mathsf{j}_i|\mathsf{q}_{i'}\rangle (C^{-1})_{i'j'} \langle \mathsf{q}_{j'}|\mathsf{j}_j\rangle. \qquad (2.18)$$

Here the inverse operator $C^{-1}$ is required to have a properly normalized projection. Using (2.9), in matrix notation the Drude weight reads

$$D = BC^{-1}B = ACA^{T}. \qquad (2.19)$$

The well known lower bound of Mazur follows in replacing in (2.18) the orthogonal projection by a smaller one.

For the Lieb-Liniger model the definition (2.16) seems to be unaccessible. But (2.19) involves only static expectations, hence a priori simpler than considering a long time limit. More details will be provided in Section 4.

The correlator $S(x,t)$ satisfies the second moment sum rule

$$\lim_{t \to \infty} \frac{1}{t^2} \int \mathrm{d}x \, x^2 \tfrac{1}{2}\big(S(x,t) + S(x,t)^{\mathrm{T}}\big) = D \qquad (2.20)$$

as a direct consequence of the conservation law, see the discussion in [44] for a particular model. Thus the Drude weight can be viewed as providing a quantitative measure on how

---

[1]It is hard to establish exactly the conditions in which the dynamics would be sufficiently mixing, but the assumption is expected, on physical grounds, to be of very wide validity.

much and how fast an initial localized perturbation is spreading ballistically. For the finer structure of the ballistic component one has to use (2.13), however.

A related quantity of interest is the time-integrated self-current correlation, where in our context "self" refers to identical reference points (say $x = 0$ by translation invariance):

$$D_{ij}^s = \int dt\, \langle j_i(0,t) j_j(0,0) \rangle_{\bar{u}}^c. \tag{2.21}$$

This is the long-time limit of the covariance matrix of the charges transferred from the left, $x < 0$, to the right, $x > 0$, halves of the system, scaled by the inverse time, which is also referred to as zero-frequency noise in mesoscopic physics [28]. We call $D^s$ simply the Drude self-weight. The diagonal entries $D_{ii}^s$, the scaled second cumulants of charge transfer, are part of the large-deviation theory for non-equilibrium transport [29, 30]. The Drude self-weight also satisfies a sum rule,

$$\lim_{t \to \infty} \frac{1}{t} \int dx |x| \tfrac{1}{2} \big( S(x,t) + S(x,t)^T \big) = D^s, \tag{2.22}$$

see [44] for a particular model.

# 3 Drude weight of the classical hard rod fluid

The material of this section has been reported already elsewhere [20]. Here the known properties are rewritten in such a way as to closely parallel our discussion of the Lieb-Liniger model. This has two advantages: The first one is more pedagogical. The underlying physics of the hard rod fluid is much simpler than the one of the $\delta$-Bose gas and it is thus easier to see how the various theory elements arise. Secondly, conjectured identities may be readily checked by using the hard rod fluid as test case.

The hard rod fluid consists of segments of length $a$ on the real line. The rods move according to their velocity until they collide, at which moment they simply exchange their velocities. Since the number of particles with given velocity is conserved, we now have an example with an infinite number of conservation laws, under the assumption that the velocity distribution is not concentrated on a finite set of $\delta$-functions. The precise definition of the fields and the equilibrium measures can be found in [2]. Here we merely follow the blue-print of Section 2. On the hydrodynamic scale the basic object is the density function $f(x,t;v)$, where the velocity $v \in \mathbb{R}$ denotes the label of the conserved field. The quantity $f(x,t;v) dx dv$ is the number of rods in the volume element $[x, x+dx] \times [v, v+dv]$, assumed to be small on the macroscopic scale, but still containing many hard rods. In approximation, the function $f$ satisfies the system of conservation laws

$$\partial_t f(v) + \partial_x \big( v_{[f]}^{\text{eff}}(v) f(v) \big) = 0, \tag{3.1}$$

which is the analogue of (2.4). The subscript $[f]$ recalls that the effective velocity $v_{[f]}^{\text{eff}}(v)$ is a nonlinear functional of $f$. Explicitly,

$$v_{[f]}^{\text{eff}}(v) = v + a(1-a\rho)^{-1} \int_{\mathbb{R}} dw\,(v-w) f(w) = v + \frac{a\rho(v-u)}{1-a\rho}, \tag{3.2}$$

which can also be written as

$$v_{[f]}^{\text{eff}}(v) = \frac{v - a\rho u}{1 - a\rho} \tag{3.3}$$

with mean density, resp. mean velocity,

$$\rho = \int_{\mathbb{R}} \mathrm{d}v\, f(v), \quad u = \rho^{-1} \int_{\mathbb{R}} \mathrm{d}v\, v f(v). \tag{3.4}$$

A generalized Gibbs ensemble (GGE) is specified by some density function $f(v)$ independent of $x$. Microscopically this means that the hard rods have uniform density and independent velocities with probability density function $\rho^{-1} f(v)$. Such background GGE is now regarded as prescribed. Test functions on velocity space are generically denoted by $\psi(v), \phi(v)$. We introduce the convolution operator

$$T\psi(v) = -a \int \mathrm{d}w\, \psi(w) \tag{3.5}$$

and the multiplication operator

$$n\psi(v) = (1 - a\rho)^{-1} f(v)\psi(v). \tag{3.6}$$

The dressing operation is defined by

$$\psi^{\mathrm{dr}} = (1 - Tn)^{-1}\psi = (1 + (1 - a\rho)Tn)\psi. \tag{3.7}$$

As we will see in Section 4, for the $\delta$-Bose gas the dressing operator is still of the form $(1 - Tn)^{-1}$, with $T$ defined through the convolution with some function $\varphi$, $T\psi(v) = (1/2\pi)\varphi * \psi(v)$. Thus Eq. (3.5) should be read as convolution with the constant function $\varphi(v) = -a$. Note that in the present case, the operator $-[(1 - a\rho)/(a\rho)]Tn$ is the projector to the constant function, and the second identity in (3.7) holds only because of this projection property.

As discussed in [20] linearizing (3.2) as $f + \epsilon\psi$ yields the linearized operator

$$A = (1 - nT)^{-1}v^{\mathrm{eff}}(1 - nT). \tag{3.8}$$

Here $v^{\mathrm{eff}}(v)$ is viewed as a multiplication operator, where for notational simplicity we dropped the subscript $[f]$. For the static covariance one obtains

$$C = (1 - nT)^{-1}f(1 - Tn)^{-1}, \tag{3.9}$$

for the current-field covariance

$$B = (1 - nT)^{-1}f v^{\mathrm{eff}}(1 - Tn)^{-1}, \tag{3.10}$$

and for the Drude weight

$$D = (1 - nT)^{-1}f(v^{\mathrm{eff}})^2(1 - Tn)^{-1}, \tag{3.11}$$

where $f(v)$ and $v^{\mathrm{eff}}(v)$ act as multiplication operators. By straightforward multiplication one notes that the relations (2.10), (2.14), (2.15), and (2.19) are satisfied.

Sometimes it is convenient to rewrite these relations as quadratic forms. For example

$$\langle \phi, C\psi \rangle = \int \mathrm{d}v\, \phi(v)f(v)\psi(v) + a(a\rho - 2)\int \mathrm{d}v\, f(v)\phi(v)\int \mathrm{d}w\, f(w)\psi(w). \tag{3.12}$$

Microscopically one would consider the stationary random field $a_\psi(x) = \sum_\ell \psi(v_\ell)\delta(x - r_\ell)$, where $r_\ell$ is the position and $v_\ell$ the velocity of the $\ell$-th hard rod. Then, as in (2.17), $C$ is the covariance

$$\langle \phi, C\psi \rangle = \langle a_\phi | a_\psi \rangle = \int \mathrm{d}x\, \langle a_\phi(x)a_\psi(0)\rangle^{\mathrm{c}}_f, \tag{3.13}$$

average in the GGE defined by $f(v)$. The first term on the right of (3.12) corresponds to the ideal gas contribution, while the second term results from the hard core repulsive potential.

# 4   The repulsive $\delta$-Bose gas

The hydrodynamic theory outlined in Section 2 is extended to the repulsive Lieb-Liniger $\delta$-Bose gas [45], which has an infinite number of conserved charges. We however keep the notation general, since with minor adaptions the main results presented are in fact valid for other integrable models of fermionic type, including the XXZ quantum spin chain and integrable relativistic quantum field theory. The corresponding results for bosonic type integrable models are also stated, see Section 6.

In second quantization the Lieb-Liniger hamiltonian is given by

$$H = \int dx \left( \tfrac{1}{2} \partial_x \hat{\psi}(x)^* \partial_x \hat{\psi}(x) + c \hat{\psi}(x)^* \hat{\psi}(x)^* \hat{\psi}(x) \hat{\psi}(x) \right) \tag{4.1}$$

with Bose field $\hat{\psi}(x)$, $x \in \mathbb{R}$, repulsive coupling constant $c > 0$, and mass of the Bose particles $m = 1$. $H$ has an infinite number of conserved charges, labeled as $\hat{Q}_j$, $j = 0, 1, \dots$. $\hat{Q}_0$ is the particle number, $\hat{Q}_1$ the total momentum, $\hat{Q}_2 = H$ the total energy, etc. The conserved charge $\hat{Q}_j$ has the density $\hat{Q}_j(x)$,

$$\hat{Q}_j = \int dx \, \hat{Q}_j(x). \tag{4.2}$$

From the conserved charges one constructs the generalized Gibbs state through

$$\rho_{\mathrm{GG}} = Z^{-1} \exp \left[ - \sum_{j \geq 0} \beta_j \hat{Q}_j \right] \tag{4.3}$$

with $\{\beta_j, j \geq 0\}$ the generalized inverse temperatures, equivalently chemical potentials. In the hydrodynamic approach the Bose gas is initialized in a local equilibrium state of the form

$$\rho_{\mathrm{LE}} = Z^{-1} \exp \left[ - \sum_{j \geq 0} \int dx \, \beta_j(x) \hat{Q}_j(x) \right], \tag{4.4}$$

assuming that the chemical potentials are slowly varying on the scale of the typical interparticle and scattering distances. Generalized hydrodynamics asserts that in approximation such structure is propagated in time according to

$$\rho_{\mathrm{LE}}(t) = e^{-iHt} \rho_{\mathrm{LE}} e^{iHt} \simeq Z^{-1} \exp \left[ - \sum_{j \geq 0} \int dx \, \beta_j(x, t) \hat{Q}_j(x) \right]. \tag{4.5}$$

The slow variation in space induces a correspondingly slow variation in time. It also means that averages of local observables at $(x, t)$ with respect to $\rho_{\mathrm{LE}}$ can be evaluated as averages with respect to $\rho_{\mathrm{GG}}$ with the properly adjusted values of the chemical potentials $\{\beta_j(x, t), j \geq 0\}$. *Remark*: For integrable lattice models, the conserved charges are written as sums over translates of local and quasi-local densities [46]. Their currents, as computed from the conservation law, have the same structure. However for the $\delta$-Bose gas our formulas are tentative. The total charges $\hat{Q}_j$ are usually defined through the Bethe eigenfunctions of $\hat{Q}_2$ by replacing the $n$-particle energy $\sum_{\ell=1}^{n} (k_\ell)^2$ simply by $\sum_{\ell=1}^{n} (k_\ell)^j$. But the corresponding local charge densities are known only up $j = 4$. We refer to [47] for a discussion. Nevertheless one would hope that, at least for appropriate conserved charges, GGE averaged densities and currents and GGE connected two-point correlation functions are still meaningfully defined. The set of appropriate conserved charges is a subtle point. There are *bona fide* GGE states for which local densities have diverging averages [48], although this does not imply divergence of their two-point correlation functions. One may restrict to the Hilbert space of pseudolocal charges, which, by

the rigorous results of [49], at least in quantum chains would be the Hilbert of functions $h(\theta)$ induced by the covariance inner product (2.17) or (2.15) (and thus, explicitly, (1.1)). Pseudolocal densities have finite integrated connected two-point functions by construction, and we expect all our results to hold for all such pseudolocal densities and their currents as long as the explicit formula gives a finite answer.

To lowest order in the variation, the family $\{\beta_j(x,t), j \geq 0\}$ satisfies a closed set of Euler-type equations, as explained in [3, 4]. We mostly follow the notation in [3]. Instead of $\{\beta_j(x,t), j \geq 0\}$ it is more instructive to write down the evolution equation in terms of the quasiparticle density $\rho_{\mathrm{p}}(x,t;\theta)$ with $\theta \in \mathbb{R}$ the label of the conserved field. The density is governed by the system of conservation laws

$$\partial_t \rho_{\mathrm{p}}(x,t;\theta) + \partial_x \left( v^{\mathrm{eff}}_{[\rho_p]}(x,t;\theta) \rho_{\mathrm{p}}(x,t;\theta) \right) = 0. \tag{4.6}$$

Comparing with (3.1), $\rho_{\mathrm{p}}(x,t;\theta)$ takes the role of the hard rod density $f(x,t;v)$. The effective velocity $v^{\mathrm{eff}}_{[\rho_p]}$ is a nonlinear functional of $\rho_{\mathrm{p}}(\cdot;\theta)$, which is local in $(x,t)$. Its precise definition will be given below. To have a more concise notation, we will mostly drop the dependence on $[\rho_p]$.

The Lieb-Liniger model has momentum $p(\theta) = \theta$ and kinetic energy $E(\theta) = \frac{1}{2}\theta^2$. As in [3], our results are valid for a general choice of $p, E$ and for future applications we retain this generality. Similarly, the higher-spin conserved charges in the Lieb-Liniger model can be chosen to have one-particle eigenvalues $h_j(\theta) = \theta^j/j!$, and our results hold for a general choice of a complete basis $h_j$ in Bethe-ansatz integrable models. In [3], the scattering amplitude is denoted by $\varphi(\theta)$, where for the Lieb-Liniger model $\varphi(\theta) = 4c/(\theta^2 + 4c^2)$. Again such specific choice is not needed in the following derivation. The operator of convolution with $\varphi$ will be denoted by

$$T\psi(\theta) = \frac{1}{2\pi} \int \mathrm{d}\alpha \, \varphi(\theta - \alpha) \psi(\alpha). \tag{4.7}$$

As for hard rods, $\phi, \psi$ are our generic symbols for smooth test functions on label space.

Let us first explain $\rho_{\mathrm{p}}$ and $v^{\mathrm{eff}}$, for which it suffices to consider the spatially homogeneous state $\rho_{\mathrm{GG}}$ with some prescribed chemical potentials $\{\beta_j, j \geq 0\}$. We define

$$w(\theta) = \sum_{j \geq 0} \beta_j h_j(\theta). \tag{4.8}$$

The quasienergies, $\varepsilon(\theta)$, are the solutions to the integral equation

$$\varepsilon(\theta) = w(\theta) - T \log(1 + \mathrm{e}^{-\varepsilon})(\theta). \tag{4.9}$$

Note that

$$\partial_{\beta_m} \varepsilon = h_m + Tn\partial_{\beta_m} \varepsilon, \tag{4.10}$$

where

$$n(\theta) = \frac{1}{1 + \mathrm{e}^{\varepsilon(\theta)}} \tag{4.11}$$

and $n$ denotes multiplication by the occupation function $n(\theta)$, that is $(n\psi)(\theta) = n(\theta)\psi(\theta)$. As before we define the dressing transformation as

$$\psi^{\mathrm{dr}} = (1 - Tn)^{-1}\psi. \tag{4.12}$$

Hence

$$\partial_{\beta_m} \varepsilon = (h_m)^{\mathrm{dr}}. \tag{4.13}$$

The quasiparticle density satisfies

$$n(\theta)^{-1}\rho_{\mathrm{p}}(\theta) = \tfrac{1}{2\pi}p'(\theta) + T\rho_{\mathrm{p}}(\theta), \quad 2\pi\rho_{\mathrm{p}}(\theta) = n(\theta)(p')^{\mathrm{dr}}(\theta). \tag{4.14}$$

Through $\rho_{\mathrm{p}}$ the average conserved charge per unit length can be computed as

$$\langle \hat{Q}_j(0) \rangle = \mathsf{q}_j = \int d\theta \, \rho_{\mathrm{p}}(\theta) h_j(\theta) = \tfrac{1}{2\pi}\int dp(\theta) n(\theta)(h_j)^{\mathrm{dr}}(\theta). \tag{4.15}$$

Here $\langle \cdot \rangle$ denotes the infinite volume GGE average (and below, in expressions such as $\langle \hat{Q}_i(x)\hat{Q}_j(0) \rangle^{\mathrm{c}}$, the superscript will again refer to the usual connected correlation functions).

Surprisingly this formalism extends also to average currents. The local current density of the $j$-th conserved charge is given through

$$\mathrm{i}[H, \hat{Q}_j(x)] + \partial_x \hat{J}_j(x) = 0 \tag{4.16}$$

and its average is [3,4]

$$\langle \hat{J}_j(0) \rangle = \mathsf{j}_j = \int d\theta \rho_{\mathrm{p}}(\theta) v^{\mathrm{eff}}(\theta) h_j(\theta) = \tfrac{1}{2\pi}\int dE(\theta) n(\theta)(h_j)^{\mathrm{dr}}(\theta) \tag{4.17}$$

with the effective velocity

$$v^{\mathrm{eff}}(\theta) = \frac{(E')^{\mathrm{dr}}(\theta)}{(p')^{\mathrm{dr}}(\theta)}. \tag{4.18}$$

*Remark*: For a well-defined dressing transformation, the operator $1 - Tn$ has to be invertible. Also, for the linear response computation in Section 5 we will need that $v^{\mathrm{eff}}(\theta)$ is strictly increasing in $\theta$ and approximately linear for large $\theta$. Such properties can be established for the Lieb-Liniger model, but more technical considerations are required which are outside this contribution.

We now extend the general relations from Section 2, valid for a finite number of conserved fields, to the Lieb-Liniger model. The charge-charge covariance matrix $C$ has to be deduced from the GGE of the $\delta$-Bose gas through

$$C_{ij} = \int dx \, \langle \hat{Q}_i(x)\hat{Q}_j(0) \rangle^{\mathrm{c}}_{\rho_{\mathrm{p}}}. \tag{4.19}$$

This quantity has been considered in [25], but our expression below seems to be new. We develop a method by which one can compute also the charge-current correlation matrix $B$,

$$B_{ij} = \int dx \, \langle \hat{Q}_i(x)\hat{J}_j(0) \rangle^{\mathrm{c}}_{\rho_{\mathrm{p}}}. \tag{4.20}$$

Then the Drude weight equals $D = BC^{-1}B$ and the linearization $A = BC^{-1}$. As a consistency check, we will also show that the so-determined $A$ agrees with linearizing (4.6) as $\rho_{\mathrm{p}} + \delta\psi$ with small $\delta$. One can also turn the logic the other way. Given the charge correlator $C$ and $A$, which in addition uses only the average currents, we compute the matrices $B, D$.

As our main result, the matrices (4.19) and (4.20) of the Lieb-Liniger model are written in a form which can be viewed as a sort of diagonalization. Thereby we arrive at a fairly explicit expression for the Drude weight. It is convenient to use the operators $T, n$ introduced above, as well as the multiplication operators $\rho_{\mathrm{p}}$ and $v^{\mathrm{eff}}$. Writing $C_{ij} = \langle h_i, Ch_j \rangle = \int d\theta \, h_i(\theta)(Ch_j)(\theta)$, and similarly for $B, D, A$ and $D^{\mathrm{s}}$, the following identities hold:

(i) *charge-charge correlator*

$$C = (1 - nT)^{-1}\rho_{\mathrm{p}}(1 - n)(1 - Tn)^{-1}, \tag{4.21}$$

(ii) *charge-current correlator*

$$B = (1 - nT)^{-1} \rho_{\mathrm{p}} (1 - n) v^{\mathrm{eff}} (1 - Tn)^{-1}, \tag{4.22}$$

(iii) *Drude weight*

$$D = (1 - nT)^{-1} \rho_{\mathrm{p}} (1 - n) (v^{\mathrm{eff}})^2 (1 - Tn)^{-1}, \tag{4.23}$$

(iv) *linearized operator*

$$A = (1 - nT)^{-1} v^{\mathrm{eff}} (1 - nT), \tag{4.24}$$

(v) *Drude self-weight*

$$D^{\mathrm{s}} = (1 - nT)^{-1} \rho_{\mathrm{p}} (1 - n) |v^{\mathrm{eff}}| (1 - Tn)^{-1}. \tag{4.25}$$

In terms of linear combinations as

$$a_\psi(x) = \sum_{j=0}^{\infty} c_j \hat{Q}_j(x), \quad \psi(\theta) = \sum_{j=0}^{\infty} c_j h_j(\theta) \tag{4.26}$$

with general coefficients $c_j$, this is

$$\langle \phi, C\psi \rangle = \int \mathrm{d}\theta \, \rho_{\mathrm{p}}(\theta)(1 - n(\theta)) \phi^{\mathrm{dr}}(\theta) \psi^{\mathrm{dr}}(\theta), \tag{4.27}$$

$$\langle \phi, B\psi \rangle = \int \mathrm{d}\theta \, \rho_{\mathrm{p}}(\theta)(1 - n(\theta)) v^{\mathrm{eff}}(\theta) \phi^{\mathrm{dr}}(\theta) \psi^{\mathrm{dr}}(\theta), \tag{4.28}$$

$$\langle \phi, D\psi \rangle = \int \mathrm{d}\theta \, \rho_{\mathrm{p}}(\theta)(1 - n(\theta)) v^{\mathrm{eff}}(\theta)^2 \phi^{\mathrm{dr}}(\theta) \psi^{\mathrm{dr}}(\theta), \tag{4.29}$$

$$\langle \phi, A\psi \rangle = \int \mathrm{d}\theta \, v^{\mathrm{eff}}(\theta) \phi^{\mathrm{dr}}(\theta)(1 - nT) \psi(\theta), \tag{4.30}$$

$$\langle \phi, D^{\mathrm{s}}\psi \rangle = \int \mathrm{d}\theta \, \rho_{\mathrm{p}}(\theta)(1 - n(\theta)) |v^{\mathrm{eff}}(\theta)| \phi^{\mathrm{dr}}(\theta) \psi^{\mathrm{dr}}(\theta). \tag{4.31}$$

For example

$$\langle \phi, C\psi \rangle = \int \mathrm{d}x \, \langle a_\phi(x) a_\psi(0) \rangle^{\mathrm{c}}_{\rho_{\mathrm{p}}} = \int \mathrm{d}\theta \, \rho_{\mathrm{p}}(\theta)(1 - n(\theta)) \phi^{\mathrm{dr}}(\theta) \psi^{\mathrm{dr}}(\theta) \tag{4.32}$$

and correspondingly for $B, D, A, D^{\mathrm{s}}$.

**Proof** of (i)-(iv): We start from the functional

$$F_g = -\frac{1}{2\pi} \int \mathrm{d}\theta \, g(\theta) \log(1 + \mathrm{e}^{-\varepsilon(\theta)}) \tag{4.33}$$

with a yet arbitrary function $g$. Then (keeping implicit the argument $\theta$ of the integrand)

$$\partial_{\beta_j} F_g = \frac{1}{2\pi} \int \mathrm{d}\theta \, g n \partial_{\beta_j} \varepsilon = \frac{1}{2\pi} \int \mathrm{d}\theta \, g n h_j^{\mathrm{dr}}, \tag{4.34}$$

where we used (4.13). With (4.15) and (4.17), we observe that the choices $g = p'$ and $g = E'$ give, respectively, the average densities and currents [3],

$$\partial_{\beta_j} F_{p'} = \mathsf{q}_j, \quad \partial_{\beta_j} F_{E'} = \mathsf{j}_j. \tag{4.35}$$

Note that $F_{p'}$ is the free energy of the GGE [24,25], and $F_{E'}$ is the "current free energy" obtained in [3] where the second relation of (4.35) was first derived.

Assume that $n$ depends smoothly on some parameter $\mu$. We take a second derivative in (4.10),

$$\partial_\mu \partial_{\beta_j}\varepsilon = T\partial_\mu(n\partial_{\beta_j}\varepsilon) = T\big(\partial_\mu n\partial_{\beta_j}\varepsilon + n\partial_\mu\partial_{\beta_j}\varepsilon\big). \tag{4.36}$$

Hence

$$\partial_\mu\partial_{\beta_j}\varepsilon = (1-Tn)^{-1}T(\partial_\mu n\partial_{\beta_j}\varepsilon). \tag{4.37}$$

Taking a second derivative also in (4.34) and combining with (4.37) yields the general relation

$$\partial_\mu\partial_{\beta_j}F_g = \frac{1}{2\pi}\int d\theta\, g^{\mathrm{dr}}\partial_\mu n\partial_{\beta_j}\varepsilon = \frac{1}{2\pi}\int d\theta\, g^{\mathrm{dr}}\partial_\mu n h_j^{\mathrm{dr}}. \tag{4.38}$$

With $\mu = \beta_i$, we find

$$\partial_{\beta_i}\partial_{\beta_j}F_g = -\frac{1}{2\pi}\int d\theta\, g^{\mathrm{dr}}n(1-n)\partial_{\beta_i}\varepsilon\partial_{\beta_j}\varepsilon. \tag{4.39}$$

We now set $\phi(\theta) = \sum_{i\geq 0}c_i h_i(\theta)$ and $\psi(\theta) = \sum_{j\geq 0}\tilde{c}_j h_j(\theta)$. Using (4.13) we arrive at the basic identity

$$\sum_{i,j\geq 0}c_i\tilde{c}_j\partial_{\beta_i}\partial_{\beta_j}F_g = -\frac{1}{2\pi}\int d\theta\, g^{\mathrm{dr}}n(1-n)\phi^{\mathrm{dr}}\psi^{\mathrm{dr}}. \tag{4.40}$$

Noting that for the choice $g = p'$, (4.35) along with (2.11) imply $C_{ij} = -\partial_{\beta_i}\partial_{\beta_j}F_{p'}$, (4.27) follows upon using the last relation in (4.14). To establish (4.22), we instead choose $g = E'$; then (4.35) and (2.11) give

$$B_{ij} = \int dx\langle\hat{Q}_i(x)\hat{J}_j(0)\rangle = -\partial_{\beta_i}\partial_{\beta_j}F_{E'}. \tag{4.41}$$

Hence our claim follows from the basic identity (4.40) together with the relations (4.14) and (4.18). Finally observing that $C^{-1} = (1-Tn)(\rho_{\mathrm{p}}(1-n))^{-1}(1-nT)$, the claims (4.29) and (4.30) are a consequence of $D = BC^{-1}B$ and $A = BC^{-1}$.

The missing piece is to reconfirm $A$ of (4.24) by linearizing the Euler type equation (4.6). We linearize the current in (4.6) as $\rho_{\mathrm{p}} + \delta\psi$,

$$\delta(v^{\mathrm{eff}}\rho_{\mathrm{p}}) = v^{\mathrm{eff}}\delta\psi + \rho_{\mathrm{p}}\delta v^{\mathrm{eff}}. \tag{4.42}$$

For $v^{\mathrm{eff}}$ we use the identity Eq. (29) in [3],

$$p'v^{\mathrm{eff}} = E' + 2\pi T(\rho_{\mathrm{p}}v^{\mathrm{eff}}) - 2\pi v^{\mathrm{eff}}T\rho_{\mathrm{p}}, \tag{4.43}$$

since $\rho_{\mathrm{p}}$ appears linearly. Then

$$v^{\mathrm{eff}} = (\tfrac{1}{2\pi}p' - T\rho_{\mathrm{p}} + M_{\rho_{\mathrm{p}}})^{-1}\tfrac{1}{2\pi}E', \tag{4.44}$$

where $M_{\rho_{\mathrm{p}}}$ is a multiplication operator by $(T\rho_{\mathrm{p}})$, acting as

$$M_{\rho_{\mathrm{p}}}\psi(\theta) = \frac{1}{2\pi}\int d\alpha\,\varphi(\theta-\alpha)\rho_{\mathrm{p}}(\alpha)\psi(\theta). \tag{4.45}$$

Variation of $\rho_{\mathrm{p}}$ yields

$$\langle\phi, v^{\mathrm{eff}}_{[\rho_{\mathrm{p}}+\delta\psi]}\rangle - \langle\phi, v^{\mathrm{eff}}_{[\rho_{\mathrm{p}}]}\rangle = \langle\phi, \rho_{\mathrm{p}}(\tfrac{1}{2\pi}p' - T\rho_{\mathrm{p}} + M_{\rho_{\mathrm{p}}})^{-1}(Tv^{\mathrm{eff}} - v^{\mathrm{eff}}T)\delta\psi\rangle. \tag{4.46}$$

Thus our task is to show that

$$(1-nT)^{-1}v^{\text{eff}}(1-nT) = v^{\text{eff}} + \rho_{\text{p}}(\tfrac{1}{2\pi}p' - T\rho_{\text{p}} + M_{\rho_{\text{p}}})^{-1}(Tv^{\text{eff}} - v^{\text{eff}}T). \quad (4.47)$$

Multiplying Eq (4.47) with $(1-nT)$ from the left yields

$$n(Tv^{\text{eff}} - v^{\text{eff}}T) = (1-nT)\rho_{\text{p}}(\tfrac{1}{2\pi}p' - T\rho_{\text{p}} + M_{\rho_{\text{p}}})^{-1}(Tv^{\text{eff}} - v^{\text{eff}}T). \quad (4.48)$$

In order to have equality, it is sufficient to show that

$$n = (1-nT)\rho_{\text{p}}(\tfrac{1}{2\pi}p' - T\rho_{\text{p}} + M_{\rho_{\text{p}}})^{-1} \quad (4.49)$$

which is equivalent to

$$n(\tfrac{1}{2\pi}p' - T\rho_{\text{p}} + M_{\rho_{\text{p}}}) = (1-nT)\rho_{\text{p}}. \quad (4.50)$$

This is satisfied because of (4.14) and we have established (4.47). ∎

There is a physically interesting consequence for the time-dependent charge-charge correlator defined through

$$\hat{S}_{ij}(k,t) = \int \mathrm{d}x\, e^{ikx} \langle \hat{Q}_i(x,t)\hat{Q}_j(0,0)\rangle^{\text{c}}_{\rho_{\text{p}}}, \quad (4.51)$$

compare with (2.13). On the hydrodynamic scale, small $k$, large $t$, $\hat{S}_{ij}(k,t)$ is approximated by

$$\hat{S}_{ij}(k,t) \simeq \langle h_i, e^{iktA}Ch_j\rangle = \int \mathrm{d}\theta\, e^{iktv^{\text{eff}}(\theta)}\rho_{\text{p}}(\theta)(1-n(\theta))(h_i)^{\text{dr}}(\theta)(h_j)^{\text{dr}}(\theta). \quad (4.52)$$

For the special case of the density, $h_i = 1, h_j = 1$, such asymptotic behavior has been derived in [27] directly from the Bethe ansatz. Here we see that the structure of the correlator holds in much greater generality.

We return to the still missing identity (4.25). There is an exact sum rule which states

$$\int \mathrm{d}x |x| \tfrac{1}{2}\big(S_{ij}(x,t) + S_{ji}(x,t)\big) = \int_0^t \mathrm{d}s \int_0^t \mathrm{d}s'\, \langle \mathsf{j}_j(0,s)\mathsf{j}_i(0,s')\rangle^{\text{c}}, \quad (4.53)$$

see [44]. Using time-stationarity on the right-hand side and the approximation (4.52) for $S_{ij}(x,t)$ on the left, one arrives at the claimed (4.25).

The hydrodynamic approximation likewise extends to the other correlation functions. Differentiating with respect to $t$ and using the conservation equations, one obtains

$$\int \mathrm{d}x\, e^{ikx}\langle \hat{J}_i(x,t)\hat{Q}_j(0,0)\rangle^{\text{c}} \simeq \int \mathrm{d}\theta\, e^{iktv^{\text{eff}}(\theta)}\rho_{\text{p}}(\theta)(1-n(\theta))v^{\text{eff}}(\theta)(h_i)^{\text{dr}}(\theta)(h_j)^{\text{dr}}(\theta). \quad (4.54)$$

Using space-time translation invariance of the averaging and further differentiating, we get

$$\int \mathrm{d}x\, e^{ikx}\langle \hat{J}_i(x,t)\hat{J}_j(0,0)\rangle^{\text{c}} \simeq \int \mathrm{d}\theta\, e^{iktv^{\text{eff}}(\theta)}\rho_{\text{p}}(\theta)(1-n(\theta))v^{\text{eff}}(\theta)^2(h_i)^{\text{dr}}(\theta)(h_j)^{\text{dr}}(\theta).$$
$$(4.55)$$

At $k=0$ one recovers the Drude weight (4.23), in agreement with its basic definition (2.16). Further, integrating (4.55) over $t \in \mathbb{R}$, the left-hand side is proportional to $\delta(k)$, since the time-integrated current is position independent because of the conservation law. Equating with the integrated right-hand side yields again (4.25). In our discussion long times means

ballistic (Eulerian) time scale, i.e. $kt = \mathcal{O}(1)$. The diffusive time scale, $t$ of order $k^{-2}$, is not covered and as an input would require some information on

$$\int_0^\infty dt \left( \int dx \langle \hat{J}_i(x,t) \hat{J}_j(0,0) \rangle^c - D_{ij} \right), \tag{4.56}$$

which currently seems to be out of reach.

Finally, we observe that, Fourier transforming (4.52) on $k$, the space-time dependent charge-charge correlator can be written in the form

$$S_{ij}(x,t) \simeq \int d\theta\, \delta(x - v^{\text{eff}}(\theta)t) \rho_{\text{p}}(\theta)(1 - n(\theta))(h_i)^{\text{dr}}(\theta)(h_j)^{\text{dr}}(\theta), \tag{4.57}$$

which has a clear physical interpretation: in the hydrodynamic limit, the correlation is built out of particles propagating, from the initial position $(0,0)$ to the position $(x,t)$, ballistically at the speeds $v^{\text{eff}}(\theta)$. The equilibrium weight is encoded in $\rho_{\text{p}}(1-n)$ and $(h_i)^{\text{dr}}$, resp. $(h_j)^{\text{dr}}$, result from the observable at the start and end point. The other correlation functions, (4.54) and (4.55), can be viewed in the corresponding way.

## 5 Linear response

### 5.1 Drude weight

The Drude weight of the Lieb-Liniger model can also be obtained from a linear response for the current. One starts from a domain wall, which means the state (4.4) with $\beta_i(x) = \beta_i - \frac{1}{2}\mu_i$ for $x < 0$, $\beta_i(x) = \beta_i + \frac{1}{2}\mu_i$ for $x > 0$, and $\beta_j = const.$ for $j \neq i$. The linear response of the $j$-th average current is defined through

$$D_{ij} = \lim_{\mu_i \to 0} \frac{\partial}{\partial \mu_i} \lim_{t \to \infty} \frac{1}{t} \int dx\, \langle \hat{J}_j(x,t) \rangle_{\mu_i}. \tag{5.1}$$

We will establish that this expression indeed agrees with (4.23).

In the context of the XXZ and Hubbard model the prescription (5.1), for the special case of charge, spin and energy currents with thermal Gibbs as reference state, is discussed in [40,41] and used for a numerical computation of the associated components of the Drude weight. These results have been combined with generalized hydrodynamics in order to evaluate exactly these components of the Drude weight [38, 39]. Earlier, a linear response formula for the Drude weight has been proposed and proved in [37, sect 6], for the diagonal case ($i = j$) and with thermal Gibbs as reference state. However instead of an initial domain wall the authors consider an initial spatially homogeneous equilibrium state and perturb the dynamics by a linear potential of the form $\mu_i \int dx\, x \hat{Q}_i(x)$. While leading to the same result, a numerical implementation seems to be more difficult when compared to the initial domain wall (5.1).

Since the right-hand side of (5.1) is evaluated at large times, we can use the asymptotic form of the resulting current, which is known to be described by a local GGE of self-similar form. We thus change the integration variable to $\xi = x/t$,

$$D_{ij} = \lim_{\mu_i \to 0} \frac{\partial}{\partial \mu_i} \int d\xi\, \langle \hat{J}_j \rangle_{\xi,\mu_i} = \lim_{\mu_i \to 0} \int d\xi\, \frac{\partial \langle \hat{J}_j \rangle_{\xi,\mu_i}}{\partial \mu_i}. \tag{5.2}$$

From [3] it is known that

$$\langle \hat{J}_j \rangle_{\xi,\mu_i} = \int d\theta\, E' n h_j^{\text{dr}}, \tag{5.3}$$

where

$$n(\theta) = n_{\mathrm{L}}(\theta)\chi(\theta > \theta_\star(\xi, \mu_i)) + n_{\mathrm{R}}(\theta)\chi(\theta < \theta_\star(\xi, \mu_i)). \tag{5.4}$$

Here $\chi = 1$, if the condition of the argument is satisfied, and $\chi = 0$ otherwise, $\theta_\star(\xi, \mu_i)$ is implicitly defined by the relation $v^{\mathrm{eff}}(\theta_\star(\xi, \mu_i)) = \xi$,

$$n_{\mathrm{L,R}}(\theta) = \frac{1}{1 + e^{\varepsilon_{\mathrm{L,R}}(\theta)}}, \tag{5.5}$$

and $\varepsilon_{\mathrm{L}}$ (resp. $\varepsilon_{\mathrm{R}}$) is determined by (4.9), where $w(\theta)$ is given by (4.8) with the replacement $\beta_i \rightsquigarrow \beta_i - \mu_i/2$ (resp. $\beta_i \rightsquigarrow \beta_i + \mu_i/2$).

Taking the derivative, the general relation (4.38) gives

$$\left.\frac{\partial \langle \hat{J}_j \rangle_{\xi, \mu_i}}{\partial \mu_i}\right|_{\mu_i=0} = \int \mathrm{d}\theta \left((E')^{\mathrm{dr}} h_j^{\mathrm{dr}} \partial_{\mu_i} n\right)_{\mu_i=0}. \tag{5.6}$$

Note that $(n_L - n_R)\delta(\theta - \theta_\star)\partial_{\mu_i}\theta_\star|_{\mu_i=0} = 0$ because $n_L = n_R$ at $\mu_i = 0$. Hence we obtain

$$\partial_{\mu_i} n(\theta)|_{\mu_i=0} = \partial_{\mu_i} n_L(\theta)|_{\mu_i=0} \chi(\theta > \theta_\star(\xi)) + \partial_{\mu_i} n_R(\theta)|_{\mu_i=0} \chi(\theta < \theta_\star(\xi)), \tag{5.7}$$

where $\theta_\star(\xi) = \theta_\star(\xi, \mu_i = 0)$. From (4.11) and (4.13),

$$\partial_{\mu_i} n_{\mathrm{L,R}}|_{\mu_i=0} = \pm\tfrac{1}{2} h_i^{\mathrm{dr}} n(1-n), \tag{5.8}$$

where $n$ is the equilibrium occupation function of the spatially homogeneous background state (4.3). Thus, inserting to the integral (5.6),

$$D_{ij} = \frac{1}{2} \int \mathrm{d}\xi \left[ \int_{\theta_\star(\xi)}^{\infty} \mathrm{d}\theta\, h_i^{\mathrm{dr}} h_j^{\mathrm{dr}} n(1-n)(E')^{\mathrm{dr}} - \int_{-\infty}^{\theta_\star(\xi)} \mathrm{d}\theta\, h_i^{\mathrm{dr}} h_j^{\mathrm{dr}} n(1-n)(E')^{\mathrm{dr}} \right]. \tag{5.9}$$

Note that the integrands do not depend on $\xi$. Let us abbreviate $g = h_i^{\mathrm{dr}} h_j^{\mathrm{dr}} n(1-n)(E')^{\mathrm{dr}}$. Then

$$D_{ij} = \frac{1}{2} \int \mathrm{d}\xi \int \mathrm{d}\theta\, g(\theta)\bigl(\chi(\{\xi < v^{\mathrm{eff}}(\theta)\}) - \chi(\{\xi > v^{\mathrm{eff}}(\theta)\})\bigr). \tag{5.10}$$

In approximation, $v^{\mathrm{eff}}$ is linear for large $|\theta|$. As can be checked for the Lieb-Liniger model, we assume that

$$\int \mathrm{d}\theta\, |g(\theta)|(1 + |\theta|^{1+\delta}) < \infty \tag{5.11}$$

for some $\delta > 0$. Then with vanishing error one can cut-off the $\xi$-integration and obtains

$$D_{ij} = \lim_{a \to \infty} \frac{1}{2} \int \mathrm{d}\theta\, g(\theta) \int_{-a}^{a} \mathrm{d}\xi \left(\chi(\{\xi < v^{\mathrm{eff}}(\theta)\}) - \chi(\{\xi > v^{\mathrm{eff}}(\theta)\})\right) = \int \mathrm{d}\theta\, g(\theta)v^{\mathrm{eff}}(\theta), \tag{5.12}$$

as claimed.

## 5.2 Drude self-weight

A linear-response formula for $D^{\mathrm{s}}$ similar to that for the Drude weight is as follows. With the same protocol as in (5.1) for the quantity $\langle \hat{J}_j(0, t) \rangle_{\mu_i}$, one writes

$$D_{ij}^{\mathrm{s}} = \lim_{\mu_i \to 0} 2\frac{\partial}{\partial \mu_i} \lim_{t \to \infty} \langle \hat{J}_j(0, t) \rangle_{\mu_i}. \tag{5.13}$$

General arguments for this relation can be given. If the GGE at $\mu_i = 0$ is an equilibrium state (that is, time-reversal symmetric), then this relation follows from standard fluctuation relations of Cohen-Gallavotti type, which can be established by general principles [30, 42, 50] (here generalized to higher conserved charges). In general, however, a GGE state is not at equilibrium, as it may carry currents. Yet it is known that all eigenstates with real eigenvalues of $\mathscr{PT}$ symmetric hamiltonians can be chosen to be $\mathscr{PT}$ symmetric. Since the Lieb-Liniger model, as well as many other integrable models, is $\mathscr{PT}$ symmetric, then its GGEs also are. A different derivation of equality (5.13) based on $\mathscr{PT}$-symmetry was presented in [51].

The equality (5.13) is very similar to the linear-response formula for the Drude weight, the difference being that the current is not space-integrated, it is the current across the origin. The calculation is similar to the one presented in the previous subsection, with the difference that we only need to evaluate all quantities at $\xi = 0$. Therefore, in (5.12) the integral over $\xi$ is replaced by the integrand at $\xi = 0$, and thus expression (5.13) coincides with (4.31).

In fact, without taking the $\mu_i = 0$ limit in (5.13), the resulting more general equality was derived under a certain property of "pure transmission" [42] (see also the derivation in [51]). This is one of a family of equalities for higher cumulants referred to as "extended fluctuation relations" [42]. The pure transmission property holds in free particle models and in conformal field theory [19], and it was conjectured in [42] to hold as well in interacting integrable models. However, we see here that this conjecture does not hold: had we not set $\mu_i = 0$ after taking the derivative, the resulting expression would not have agreed with (4.31). The term proportional to $\delta(\theta - \theta_\star)$ discussed just after (5.6) does not contribute at $\xi = 0$ even with $\mu_i \neq 0$, because at $\xi = 0$ we have $v^{\text{eff}}(\theta_\star) = 0$, thus $(E')^{\text{dr}} = 0$. However, keeping $\mu_i$ nonzero, we have instead of (5.8) the relation $\partial_{\mu_i} n_{\text{L,R}} = \pm \frac{1}{2}(h_i)^{\text{dr}}_{[n_{L,R}]} n_{L,R}(1 - n_{L,R})$, where the index indicates that the dressing operation is with respect to the left (right) bath $n_L(\theta)$ $(n_R(\theta))$. Therefore we obtain (5.9), again without $\xi$ integration and instead at $\xi = 0$, but where in the first $\theta$-integral $h_i^{\text{dr}}$ is replaced by $(h_i)^{\text{dr}}_{[n_L]}$, and in the second integral, by $(h_i)^{\text{dr}}_{[n_R]}$. The resulting expression is therefore different from (4.31). It would be interesting to understand more at length the consequences of the lack of pure transmission, and the general arguments for (5.13) and related equalities for higher cumulants.

# 6 Discussion

There are a number of immediate generalizations to the above results. First, as mentioned in the introduction, the results (4.27) - (4.30) are expected to hold in Bethe-ansatz integrable models of fermionic type. In general, with multiple species of particles, $\theta$ stands for a multi-index, involving both the velocity (or the quasi-momentum) and the particle type, and integrals over $\theta$ include sums over particle types; see e.g. [3, 4, 11]. Second, the (generalized) thermo-dynamic Bethe ansatz was also developed for models with bosonic statistics, see e.g. [24]. In this case, (4.9) is replaced by

$$\varepsilon(\theta) = w(\theta) + T \log(1 - e^{-\varepsilon})(\theta) \tag{6.1}$$

and the occupation function is

$$n(\theta) = \frac{1}{e^{\varepsilon(\theta)} - 1}. \tag{6.2}$$

The dressing operation and the values for averages are otherwise of the same form. We use $-\partial_{\beta_j} n = n(1 + n)\partial_{\beta_j}\varepsilon$ to repeat our computation from before and find that (4.21) - (4.25) remain valid provided $\rho_{\text{p}}(1 - n)$ is replaced by $\rho_{\text{p}}(1 + n)$. The results are those expressed in (1.1) - (1.4) with $\sigma = -1$. Third, it should also be possible to generalize to classical soliton-like gases [16], taking inspiration from the hard rod model, although a precise discussion

of this is beyond the scope of this paper. The expected results are those obtained using the classical (Boltzmann) occupation function $n(\theta) = e^{-\varepsilon(\theta)}$, giving (1.1) - (1.4) with $\sigma = 0$. The factors $1 - n$ (fermions), $1 + n$ (bosons) and 1 (classical particles) represent the effect of the statistics of the fundamental components of the gas. Correlations are reduced in the fermionic case when the occupation is larger because of the Fermi exclusion principle. On the contrary, bosons display a condensation effect, increasing correlations; while classical particles are not subject to any nontrivial statistics. For classical integrable field theory, radiative components may give rise to occupation functions with Rayleigh-Jeans form. We hope to present complete derivations in a future work.

Some of the techniques introduced here should generalize to other space-time phenomena on the Euler scale. For instance, since no entropy is produced, the general rule states that correlations are governed by the linearized Euler equations. If the initial state has spatial variations, then the linearization is with respect to a space-time dependent background, and one could write down the equation (2.12) with space-time dependent linearization matrix $A$. Another possibility that can be accessed similarly is to have external potentials varying on the Euler scale. We leave for future works the analysis of such equations and of their solutions.

# Acknowledgments

BD is grateful to A. Bastianello, T. Prosen, T. Yoshimura and G. Watts for discussions, and especially to T. Yoshimura for pointing out an argument that is used in subsection 5.2. We thank X. Zotos for email discussions and comments on the first version of this paper.

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
