# Peer review of "Drude Weight for the Lieb-Liniger Bose Gas"

_SciPost Physics, doi:SciPost Phys. 3, 039 (2017)_

## Round 2 · Referee Report · Anonymous (Referee 1) · 2017-8-6

Strengths

1- Drude weights for interacting integrable quantum systems are neatly discussed within the universal theory of linear hydrodynamics.

Weaknesses

1- The authors provide no quantitative analysis and/or numerical evaluations of their formulae.

Report

Many recent studies have been devoted to studying various aspects of anomalous transport properties in exactly solvable quantum interacting lattice systems and field theories. In particular, it has been established that the late-time evolution of such systems on large scales is accurately captured by the generalized hydrodynamics. In the present work, the authors linearize the hydrodynamic equation of motion and show how (aside from an infinite number of local conserved fields) integrable quantum models display many formal similarities to the conventional hydrodynamic theory of classical interacting particle fluids. To uncover this analogy, the authors briefly revisit a system of elastically interacting hard rods.

The work concentrates on the Lieb-Liniger gas of interacting bosons, a model which has received a great deal of attention in the past years. The authors provide present explicit expressions for generic Drude weights, the so-called Drude self-weights, and also derive the form of static charge-charge and charge-current correlations. The results are nicely cast in terms of equilibrium state functions which are accessible and efficiently computable with the Thermodynamic Bethe Ansatz method. Furthermore, by linearizing the hydrodynamic equation of motion, the authors also give an expression for the dynamical structure factor valid at long wavelengths and low frequencies, thereby confirming the recent results found with the form factor approach.

In my opinion, this is a well-written work of high pedagogical value. Despite many of the formal considerations presented by the authors follow straightforwardly from the standard hydrodynamic theory, the manuscript still contains a handful of important new contributions which include (but are not limited to) compact analytic formulae for generic Drude (self-)weights and generalized susceptibilities, linearized evolution operator and the low-k low-frequency structure factor. There is little doubt that these contributions will improve conceptual understanding and show useful in future applications.

Before recommending this work for publication, I invite the authors to address the comments below and offer extra clarifications on a few loose points.

Requested changes

1- Given that the work is primarily concerned with the local conserved fields and equilibrium states in exactly solvable interacting models, I think that it would be valuable for the readers to include (besides review articles [6-8] are about non-interacting particles) additional references which are devoted to generalized Gibbs ensembles in interacting theories (e.g. in the Heisenberg spin chain).

2- It is stated that "Our method and expression are how ever new. Formula (1.3) generalizes the early results [27, 28], and as a consistency check, we show in Section 5 that it is reproduced in complete generality by the linear response calculation, thus fully confirming these early results."

I had hard time understanding in what sense (1.3) generalizes the earlier computations of [27,28], where spin and charge Drude weights are evaluated by employing Kohn's formula (i.e. energy levels curvature in the presence of external gauge potentials). It is not obvious from the arguments presented in the manuscript how the linearized hydrodynamics approach (which is based on the properties of excitations) relates to the Kohn's formulations (which uses finite-size correction to the spectrum).

3- I did not find the requirements that "dynamics [should] be sufficiently random" (p.5) or "[that the] dynamics is sufficiently mixing" (p.7) very informative or precise enough to convey a clear meaning. What are these extra assumptions about and how serious (or relevant) they are in practice? Is it conceivable that the so-called "non-sufficiently mixing" dynamics occurs in the Lieb-Liniger model and eventually makes the whole analysis on the hydrodynamic scale inapplicable? My recommendation would be to either provide some additional information (and references) or simply suppress these technical matters for the sake of clarity (unless they prove to be vital).

4- "For integrable lattice models, the conserved charges are written as a sum over translates of a strictly local operator."

By assuming that strictly local operators refer to operators whose densities are supported on a compact region in space, then the statement is not accurate. There has been a lot of activity on the notion of locality in statistical mechanics (primarily quantum lattice models) concluding that conservation laws which enter in a local statistical ensemble a-la Eq.(4.3) have typically quasi-localized (i.e. exponentially-decaying with the distance) densities. Various applications and aspect have been reviewed in [J. Stat. Mech.: Th. and Exp., 2016(6), p.064008].

5- "However for the δ-Bose gas our formulas are wishful thinking..." and "Similarly, the higher-spin conserved charges in the Lieb-Liniger model can be chosen to have one-particle eigenvalues $h_{j}(θ)=θ^{j}/j!$, and our results hold for a general choice of a complete basis $h_{j}$ in Bethe-ansatz integrable models"

Regarding this subtle behaviour, I would like mention ref. [PRA, 89(3), p.033601], where it is demonstrated that there exist certain local equilibrium states in the Lieb-Liniger model in which most of the conserved quantities $Q_j$ (as defined by the authors) become singular. Nonetheless, in my understanding, the functions $w(\theta)$ which appear in Eq.(4.8) always remain well-defined physical quantity.

6- On the dressing transformation introduced in Eq.(4.12): As far as I know, the dressing transformation in Bethe Ansatz expresses energy (or charge) shifts which an excited state experience after adding or removing particle-hole excitations (and is defined in terms of the so-called shift function). On the other hand, the authors operate with a simpler "dressing kernel" which is given by Eq.(4.13) and acts on the derivatives of bare (rapidity-dependent) quantities. Since this can be confusing to some readers, I hope that the authors can clarify this.

---

## Round 2 · Referee Report · Anonymous (Referee 2) · 2017-8-31

Strengths

1- Explicit expressions are obtained for the connected correlations of charges and currents in an interacting integrable model

Weaknesses

1- The paper is not self-contained 2- Some steps of the proof of the main results are not clear

Report

In this paper the authors disclose four identities for the connected correlations of conserved charges and relative currents, in the repulsive Lieb-Liniger model.
In particular, they obtain expressions in terms of thermodynamics-Bethe-ansatz quantities. The authors compute the Drude weight for the entire set of charges and discuss generalizations to other interacting integrable systems, including models with bosonic statistics.

In my opinion, the results reported in this paper are important, especially for their supposed generality. If the derivation of the identities were clear, I would have recommended this paper for publication. There are however some fundamental steps, in the main derivation, that I fail to understand. Thus, I prefer to defer my recommendations until the authors will have cleared up my doubts.

I’m referring to the proof of (i)-(iv), which starts at page 14. The authors introduce a functional that depends on an arbitrary function. The physical meaning of such functional is not explained, and the authors do not establish any connection with actual thermodynamic quantities.
I am aware that that functional can be used to compute the expectation values of charges and currents, and I have been able to follow the proof until eq. (4.40).
Then, I do not see what arguments the authors are using to connect the second derivatives of the functional to the connected correlations.
Such step could be justified by establishing a connection between the functional and a partition function. In one case (for the matrix C), I know that this is possible.
For the correlations involving also the currents, I doubt that such connection has been already established: as far as I know, the currents are not known as operators (and their expectation values are known only if computed in stationary states).
Perhaps the authors are using a different argument or some results that I do not know. If so, the authors should guide the reader through the relevant literature, possibly providing a sketch of the main steps of the proof.

Requested changes

1- The proof of (i)-(iv) must be improved. 2- I think that the discussion below eq. (1.4) can be misunderstood. Indeed, the authors present their dressing operation as “standard”, but, from my point of view, it is not. In particular, it is different from the dressing operation considered in standard textbooks like “Quantum Inverse Scattering Method and Correlation Functions”, by Korepin, Bogoliubov, and Izergin. 3- The last sentence of page 4 must be rewritten. 4- In (2.1), also the current is time dependent. 5- In (2.4), the right hand side of the equation is missing. 6- The notations introduced above (2.6) are not consistent with the rest. In particular, $\delta\phi$ becomes $\vec \phi$. 7- I suggest the authors to include a relevant reference before (2.18). 8- What is the basis of the matrices in (4.21)-(4.25)? (Where are $h_j$?) 9- I wonder whether there is some implicit assumption behind (5.2), in particular on the form of the corrections to generalized hydrodynamics.

---

## Round 2 · Referee Report · Anonymous (Referee 3) · 2017-9-4

Strengths

1- Exact results about out-of-equilibrium interacting models, relevant for experiments 2- General conclusions holding for a large class of integrable models 3- Application of generalized hydrodynamics to the computation of correlation functions in integrable models

Weaknesses

1- The work assumes some knowledge about hydrodynamics and integrable models without giving clear references, so it is hard to read for someone not exactly in these fields 2- Most of the proofs are just sketched (in particular section 2 and 3) without sufficient hints. 3- The limits of validity of the current formalism in the hydrodynamic regime is not discussed.

Report

I think that this is a high-quality work which contains very interesting results. I appreciate the effort made by the authors in trying to remain very general and have conclusions which apply to a large class of models. Moreover, the authors manage to obtain neat results about correlation functions, which match and generalize those obtained by form factors with a much more technical approach.

However, in my opinion, the clarity of the paper can and should be improved, as many points are discussed assuming a deep knowledge of the field by the reader.

Requested changes

1- In the introduction Euler equations and Navier-Stokes equations are mentioned as expansions of spatial-gradients at different orders. However the discussion is very sketchy and no reference is provided. 2) In the general discussion of Sec. 2, it is never clarified if the authors refer to a classical or quantum model (or to both). In the introduction, it is said that the validity for "soliton-like gases" is conjectured, but then this quantum to classical limit is never discussed again. Also in Sec. 3 it should be clarified that this is a classical model. 3) The discussion below eq. 2.1 could be made clearer. In particular, it would be useful to explain clearly why the space of stationary states has the same dimension as the number of conserved quantities. Moreover, the term "stationary stochastic dynamics" is used without clarifying at all the origin of this stochasticity. 3) "truncated" below eq. 2.2 is not very intuitive: "connected" would look like a much more standard terminology in statistical physics (it would also be consistent with the $c$ subscript). 4) The notation $\delta\phi$ around eq. 2.6 is not coherent (it is not a vector and the $\delta$ is dropped just below). 5) Eq. 2.10 should be discussed more explicitly ("by the chain rule" seems a rather poor explanation). 6) Also Eq. 2.11 could be made more clear linking it explicitly to 2.6; this would also allow clarifying its validity. 7) Below Eq. 2.15: it would be useful to define at least once the meaning of "stationary in x". 8) The derivation of Eq. 3.8 (though correct) is completely left to the reader, without any hint. 9) $\chi$ is never defined in Eq. 5.4. 10) The discussion in Sec. 5.2 about the validity (or violation) of 5.13 without the limit $\mu \to 0$ is extremely implicit and not self-contained. 11) The authors should clarify better that hydrodynamics is essentially used as a tool to compute correlation functions in an homogeneous state. Large-distance correlation functions for inhomogeneous state do not seem to be accessible within this framework. This is only discussed in the conclusions, where the "Euler scale" is mentioned, without any further clarification. 12) It would be interesting to see a discussion about why $f$ in Eq. (3.9) (and below) is replaced by $\rho_p(1-n)$ in Eq. 4.21.

---

## Round 3 · Referee Report · Anonymous (Referee 1) · 2017-10-17

Report

The authors have revised their manuscript according to the suggestions, added some clarifications and made several minor improvements in the text. I do not have any additional relevant remarks and thus recommend the manuscript for publication.

Requested changes

No changes requested.

---

## Round 3 · Referee Report · Anonymous (Referee 2) · 2017-11-22

Strengths

1- Explicit expressions are obtained for the connected correlations of charges and currents in an interacting integrable model

Weaknesses

1- There are assumptions that have not been checked.

Report

Although the assumptions made to obtain the main results have not been checked, they sound reasonable.
I recommend this paper for publication in the present form.

---

## Round 3 · Author Response

We thank the three referees for their careful reading of the manuscript and for their comments.

{\bf Referee 1}

1) Reviews [8-10] (new numbering) are in fact not about non-interacting particles. In particular, the Heisenberg chain is actually dealt with in review [9], see their section VII called ``Relaxation in interacting integrable models" for instance. We believe it is sufficient to put these reviews as references -- there is a huge amount of specific references about many models (Heisenberg chain and others), but none is of direct use here, as our formalism is solely based on the thermodynamic Bethe ansatz.

2) We agree that the method of Kohn's formula and the hydrodynamic methods are very different, and we do not wish to comment on how these methods may or may not be related. In fact we do not claim to generalize the {\em method} used, rather we simply claim to confirm and generalize the {\em results} (``Formula 1.3 generalizes the early results..."), independently of the method used. The results of [31,32] (new numbering) were about spin and charge Drude weight. Here we note that our formula is the same as in [31,32] when specialized to spin and charge in the Heisenberg chain, and we generalize the result to all conserved currents. This thus confirms and generalizes the results of [31,32]. In addition, as a further confirmation, as said in the text, in [38,39] GHD was combined with a linear-response formula to calculate numerically the Drude weight, and the result were confirmed by numerics. However in [38,39] there was no explicit formula for the Drude weight. In Section 5 we show that the linear-response technique reproduces our formula. Thus this shows explicitly that the numerical results of [38,39] confirm the early results of [31,32]. We have adjusted the last sentences of the paragraph to clarify this.

3) It is hard to characterize precisely in general the conditions in which hydrodynamic (i.e. sufficient mixing) applies, but hydrodynamics is expected to have wide validity. We have deleted the first occurrence, but we kept the second and added a footnote, as we believe it is important at least to express what is expected about sufficient mixing.

4) Indeed we are aware of this fact (see e.g. [Thermalization and pseudolocality in extended quantum systems, Commun. Math. Phys. 351, 155 (2017)] by one of us), but wanted for simplicity to avoid this subtlety. However the referee is right to point this out, and we shouldn't have avoided the subject. We have added a note on this (pp 11-12) with a reference to the review suggested, and we have extended the paragraph there (see point 5 below).

5) Yes this is true, thank you for pointing this out. Well-defined GGE states $w(\theta)$ giving rise to singular averages of conserved densities do indeed occur in quenches. Our formalism is expected to apply to all GGE states, including these. We make two comments. First, finiteness or not the average of a density in a GGE, is not directly related to finiteness or not of two-point functions (1.1)-(1.4) involving it. It depends on the ways $\rho_{\rm p}(\theta)$ and $h(\theta)$ behave at large $|\theta|$. Second, given a state, the space of pseudolocal operators is exactly the Hilbert space induced by the inner product (2.17). This fact was proven in quantum chains in [Thermalization and pseudolocality in extended quantum systems, Commun. Math. Phys. 351, 155 (2017)], and is expected as well in the Lieb-Liniger model. Restricting to conserved pseudolocal charges, this is therefore the Hilbert space of functions $h(\theta)$ induced by the inner product (1.1). Thus in this space all integrated two-point functions are finite, by construction. Any conserved quantity whose modulus with respect to (1.1) is infinite, would simply not be part of this space -- hence is not a bona fide local (or quasi-local) conserved charge. We expect our results to hold for all such pseudolocal densities, and, for formulae (1.2)-(1.4), under the additional condition that they give finite answers. We have adjusted the paragraph at the top of page 12 to account for this discussion.

6) There are many ways of expressing dressed quantities. Here, since we are dealing with a field theory, we follow: (i) the paper [24] where the dressing transformation as we use here was derived in integrable QFT, and (ii) the paper [3] where the effective velocity (4.18), with bare quantities being rapidity-differentiated and then dressed, was derived, in particular in the LL model considered here. It seems to us that going into the details of the various dressing operations in TBA is a bit beyond the scope of this paper: we simply want to use the results already derived and apply them to calculate new quantities.

{\bf Referee 2}

1) We tried to improve, but we are not completely sure about the difficulties.

First of all we work with a fixed GGE, which is stationary and spatially homogeneous, which is stated clearly in the paper. Thus we can use all results known in stationary states, including the averages of currents. Starting from the basic identities, we compute derivatives in $\beta_j$. The functional $F_g$ introduced does not have physical meaning for arbitrary $g$. This is a tool for the computation, and its derivative in $\beta_j$ is evaluated using (4.13), as mentioned in the text. It has physical meanings for $g=p'$ (the generalized free energy) and $g=E'$ (the generalized ``current free energy''), something which can be found in [3], but which is not necessary here for the derivation. We have nevertheless added a comment just after (4.35). The relations (4.35) were explained and derived in [3], here we just remind them.

The fact the derivative in $\beta_j$ of any one-point average is given by a spatial integral of the connected correlation with the $j^{\rm th}$ conserved density, is a standard property of canonical ensembles, in particular of GGEs. Then, by (4.35) the second derivative of $F_{p'}$ is the two-point connected correlation of conserved densities, thus $C_{ij}$. Similarly, by (4.35) again the second derivative of $F_{E'}$ is the two-point connected correlation of one conserved current with one conserved density, thus $B_{ij}$. Nothing else is needed. Note that the fact that $F_{p'}$, $F_{E'}$ and certain ``free energies" is fully encoded in (4.35).

For Lieb-Liniger, the conserved charges are conjectured from the Bethe ansatz, but it is true that the local currents have never been defined properly as local operators in terms of the fundamental fields of the model. Nevertheless, we do not need any explicit expressions of currents here. In fact, by the statement that a charge is local, by general principles of QFT it is expected that it has an associated local conservation law, thus a local density and a local current. In [3] averages of such local currents were derived in the Lieb-Liniger model, using the non-relativistic limit of the sinh-Gordon model and a derivation in the latter model that makes use of crossing symmetry of relativistic QFT. But this explanation is not needed for the derivation: only the results on the average currents (4.17) are needed, recalled in the text. We do not think it is appropriate here to recall the full argument of [3], as it is beyond the scope.

2) The dressing operation is one that is used extensively in the paper [24] for instance (although it is not explicitly called ``dressing" there), in the context of the thermodynamic Bethe ansatz; this is the reference cited in this paragraph. We believe this paper is relatively standard; this dressing operation has been used in other TBA papers afterwards based on [24], and most importantly, it was used in the work [3] on which the present paper is based. We have added the explicit references just after the sentence for more precision, but it would seem inappropriate to start discussing in the present paper the various dressing operations used in various references, given that it is based on the formalism and notations of [3].

3) done

4) corrected.

5) corrected.

6) corrected.

7) done

8) These are operator identities, which can be implemented in any basis. The most natural basis is given in (4.27) - (4.32). Then all operators are seen as acting on functions of $\theta$; they are in general integral operators (such as $T$), or multiplication operators (such as $n$, $\rho_{\rm p}$ and $v^{\rm eff}$). This is explained for instance in (4.7) and just below (4.11). Also, as defined in the paragraph above (4.7), $h_j(\theta)=\theta^j/j!$. We have added a sentence just before (4.21) in order to further clarify.

9) The only assumption is that there is a self-similar solution depending on $x/t$. As far as we understand, nothing special needs to be assumed about the corrections to GHD.

{\bf Referee 3}

1) reference provided and discussion improved.

2) We have adjusted section 2. We say explicitly classical, quantum, stochastic, and state classical hard rods in section 3. For soliton-like gases, this is conjectured, and we do not wish to provide further discussion as this would require another paper.

3) This has been clarified. Note that most of these are relatively standard concepts, and do not need much specificity on the underlying dynamics. See the book [1], or even the work [2] where explanations are given.

3bis) corrected

4) corrected

5) done

6) done

7) done

8) We refer here to the DS paper on hard rods [20] where this identity is written in a slightly different form. Also, the $A$ operator is fully derived in the case of the Lieb-Liniger model in section 4, and as we hope should be relatively clear from the text (see (3.5) and (4.7)), the specialization $\varphi$ to a constant reproduces the hard-rod case.

9) done

10) All references are provided, and it would be much beyond the scope of this paper to go into any detail of ``extended fluctuation relations" and related concepts. What is the meaning of not being self-contained here? In any case we have tried to clarify the discussion, making the steps more explicit.

11) We have clarified the discussion, and have stated in the introduction, see page 3, that the results apply to homogeneous, stationary states. Large-distance correlation functions in inhomogeneous states are in fact accessible, but beyond the scope of this work.

12) Comments have been made in section Discussion.

---

## Round 3 · List of Changes

• References added: [1,22,23,46,48,49]
  • Introduction improved
  • Discussion in section 2 improved, in particular eq 2.12 and footnote 7 added
  • Remark concerning GGEs expanded on pp 11-12
  • Explanations improved on pp 14-16
  • Explanations improved in subsection 5.2, p 21
  • Discussion expanded pp 21-22.

---

## Editorial Decision

published